# Isolation, Identification of Carotenoid-Producing *Rhodotorula* sp. from Marine Environment and Optimization for Carotenoid Production

**DOI:** 10.3390/md17030161

**Published:** 2019-03-08

**Authors:** Yanchen Zhao, Liyun Guo, Yu Xia, Xiyi Zhuang, Weihua Chu

**Affiliations:** 1Department of Pharmaceutical Microbiology, School of Life Science and Technology, China Pharmaceutical University, Nanjing 210009, China; 17712936602@163.com; 2Department of Microbiology, Nanjing Institute of Fisheries Science, Nanjing 210036, China; lyguo801@163.com; 3Bureau of Ocean and Fisheries of Jiangsu Province, Nanjing 210003, China; xiayu6858@163.com

**Keywords:** carotenoids, optimization, red yeast, *Rhodotorula* sp.

## Abstract

Carotenoids are natural pigments found in plants and microorganisms. These important nutrients play significant roles in animal health. In contrast to plant production, the advantages of microbial fermentation of carotenoids are the lower media costs, fast growth rate of microorganisms, and the ease of culture condition control. In this study, a colony of red pigment-producing yeast, *Rhodotorula* sp. RY1801, was isolated from the sediment of marine environment with the potential to produce carotenoids. Optimization of carotenoid production in *Rhodotorula* sp. RY1801 was also discussed. The optimum conditions found for carotenoid production were as follows: temperature, 28 °C; pH 5.0; carbon source, 10 g/L glucose, nitrogen source, 10 g/L yeast extract, maximum concentration of 987 µg/L of total carotenoids was obtained. The results of this study show that the isolated yeast strain *Rhodotorula* sp. RY1801 can potentially be used in future as a promising microorganism for the commercial production of carotenoids.

## 1. Introduction

Carotenoids are pigments that exist in a wide variety of plants and microorganisms. They are characterized by yellow, orange, red or purple coloration [1]. Carotenoids have been proven to play important roles in animal health as precursors of vitamin A, scavengers of active oxygen, and enhancers of in vitro antibody production. Therefore, they are widely applied in animal feed additives as nutrient supplements, food, pharmaceutical, and cosmetic industries as dyes/colorants and functional ingredients [2]. Carotenoids are in high demand throughout the world, so a suitable method for an industrial production of carotenoids producing is needed. Most of the carotenoids are extracted from plants like annatto, tomato, grapes, carrot, paprika, etc. Carotenoids can also be produced from microorganisms [3].

Carotenoids can be produced by numerous microorganisms. Filamentous fungi, yeasts, bacteria and algae, such as *Streptomyces chrestomyceticus*, *Blakeslea trispora*, *Phycomyces blakesleeanus*, *Flavobacterium* sp., *Phaffia* sp., and *Rhodotorula* sp., Actinomycetes have been described as carotenoid-producing microorganisms [4,5]. The production of carotenoids from microorganisms have advantages over plants, such as higher yields, less batch-to-batch variations, easily manipulated, and no seasonal or geographic variations [6,7]. Carotenoid producing microorganisms, such as bacteria and archaea, algae and fungi; are abundant in the natural environment. Microalgae are currently the main sources of industrial carotenoid production [8], but other microorganisms could be valid alternatives [9]. With the rising demand of carotenoids, there has been renewed interest in identifying novel carotenoid-producing microorganisms. Yeast is the most suitable candidate for carotenoid production because of its fast growth rate, and the ease of cultivation. Yeast has the potential to produce large amounts of carotenoids such as lycopene, β-carotene, astaxanthin, torulene and torularhodin, etc. Carotenoid-producing yeasts are mainly represented by the genera *Rhodotorula* sp., *Rhodosporidium* sp., *Sporobolomyces* sp., *Xanthophylomyces* sp. [10,11]. Microorganisms that inhabit marine environments have been considered useful natural sources for new biomolecules production. Marine microorganisms possess unique metabolic and physiological features. They have evolved protective mechanisms compared to terrestrial microorganisms, which include the accumulation of bioactive compounds. It is also considered that the production of these bioactive compounds may be relatively easy by marine microorganisms [12,13]. The aim of the current study was carried out to isolate, identify carotenoid-producing strains from the marine environment, and optimize the nutritional and environmental parameters for their carotenoid production.

## 2. Results and Discussion

### 2.1. The Isolation and Identification of Carotenoids Producing Yeasts

A total of six morphologically distinct yeasts with red pigment were isolated from marine sediment samples as pure cultures (designated as RY1801–RY1806). Among the isolates, only the strain RY1801 had rapid growth and high pigment producing abilities, which was subsequently used for further study. The isolate RY1801 developed mucous, smooth surface and red-colored colonies on YPD agar plate (Figure 1A), and the growth was frequently observed in the microscopic examination (Figure 1B). Cells of the isolated strain RY1801 had an oval shape, the RY1801 cells size was 4–6.5 μm × 2–3.5 μm and had a colony diameter of 1.5 mm after 24 h cultivation (Figure 1B). The liquid medium changed to red after 24 h cultivation (Figure 1C). It has assimilated sugars such as glucose, galactose, sucrose, maltose, melezitose, and raffinose. The nitrate assimilation was positive. Further biochemical tests were carried out and are listed in Table 1. These biochemical results were not sufficient for classification to the genus, so the Internal Transcribed Spacers (ITS) of the ribosomal DNA sequence was amplified.

The ITS sequence obtained from strain *Rhodotorula* sp. RY1801 (GeneBank MH760806) was compared with the sequences in the GeneBank database and revealed that the strain RY1801 had 99% homology to *Rhodotorula babjevae*. The nucleotide sequence of the ITS region from RY1801 strain was identical to two other *R. babjevae* sequences included in the phylogenetic tree (Figure 2). Based on morphological, physiological characteristics, and ITS sequence, the isolated red yeast RY1801was tentatively named R. babjevae RY1801 and deposited at the China General Microbiological Culture Collection Center (CGMCC, Beijing) as CGMCC No. 15980.

Several yeast species can synthesize carotenoids, in particular, the genera *Xanthophyllomyces*, *Rhodotorula*, *Sporobolomyces*, and *Phaffia* have been used to produce carotenoids [14,15]. The production of carotenoid pigments in numerous natural isolates of the genera *Rhodotorula* has been studied by others, such as *Rhodotorula glutinis*, *Rhodotorula minuta*, *Rhodotorula mucilaginosa*, *Rhodotorula acheniorum* and *Rhodotorula graminis* [16]. El-Banna et al. isolated 46 yeast isolates from natural environments, all the strains belonged to *Rhodotorula glutinis* [17]. Muthezhilan et al. isolated a marine yeast *Rhodotorula* Sp. (Amby109) which can produce carotenoid pigments [18]. In this study, we isolated and identified the marine yeast which can produce red pigment, based on morphological, physiological characteristics and ITS sequence, and our results showed that the isolated red yeast RY1801 belonged to *Rhodotorula* sp.

The carotenoid pigments extracted from *Rhodotorula* sp. RY1801 have shown no inhibitory activity against all the detected strains. Muthezhilan et al. results showed that the pigment derived from marine yeast *Rhodotorula* Sp. (Amby109) have strong antimicrobial activity [18].

### 2.2. Effects of Various Parameters on Biomass Growth and Carotenoids Production

#### 2.2.1. The Incubation Temperature

Incubation temperature ranging from 20 to 37 °C were checked for the biomass and carotenoids production in *Rhodotorula* sp. RY1801. As shown in Figure 3, the optimal temperature for biomass and carotenoids production was 28 °C, although reduced biomass and carotenoid production was seen and tested at other temperatures. Our results were similar to the results provided by others. Other studies revealed the optimal temperature for maximum *Rhodotorula glutinis* growth and carotenoids production was 29 and 30 °C, respectively [19] in monoculture and 30 °C in co-culture with lactic acid bacteria [20]. The temperature also has an effect on the regulation of enzymes involved in carotenoids production [21].

#### 2.2.2. Culture Medium pH

The influence of culture medium pH on biomass growth and carotenoids production in *Rhodotorula* sp. RY1801 was evaluated in YPD medium at 28 °C. As seen in Figure 4, the optimal initial pH under our culture conditions was pH 5 and similar biomass and carotenoids concentrations were seen at pH 6.0 and 7.0. Our results were similar to those by other workers. A study by Latha et al. indicated that the *R. glutinis* biomass increased as the initial culture pH increased from 5.5 to 7.5, although optimal carotenoids production was pH 5.5 [22]. A similar optimal pH 5.5 was observed for β-Carotene production in a related species, *Rhodotorula acheniorum* [23].

#### 2.2.3. Carbon Sources

Carbon has been considered an important source for the energy supply and growth of microorganisms and is widely studied in the context of microbial fermentations. We investigated the influence of several carbon sources on biomass, and carotenoid production under culture conditions with initial pH 5.0 at 28 °C. Among the several carbon sources tested, glucose proved to be the most suitable carbon source for carotenoid production, with 962 µg/L of carotenoid (Figure 5). This may be due to the fact that glucose can easily be assimilated in the metabolic pathway for biosynthesis of carotenoids. The type of carbon source has a significant influence on carotenoids production, and their effects may differ depending on the yeast strains [24].

#### 2.2.4. Inorganic Nitrogen and Organic Nitrogen Sources

The influence of different nitrogen sources on biomass and carotenoids production was investigated with culture media containing 2% (*w/v*) glucose, initial pH 5.0, at 28 °C. Among the tested nitrogen sources, yeast extract was proved to be the most suitable nitrogen source for carotenoids production, with 987 µg/L of carotenoid (Figure 6). The influence of nitrogen sources on carotenoids production in *Rhodotorula* sp., also depend on the different strains.

Optimization of cultural conditions is necessary in microbial fermentations for carotenoids to fully exploit the potential of selected microbial strain. The fermentation conditions for the production of carotenoids by the new isolated *Rhodotorula* sp. strain was optimized in shake flasks. With different culture conditions, the amount of biomass varied widely from 3.21 to 5.63 g/L and the total carotenoids content varied from 589 to 987 µg/L. The biomass yield in our study is lower than others, which could be attributed to the short time of culture (3 days). El-Banna et al. found that *Rhodotorula glutinis* strain NO. 0 produced 7 g/L dry biomass and 266 μg/g cellular carotenoids, 1.6 μg/L volumetric carotenoids after growing at 30 °C for 4 days [16]. Hamidid et al. have also reported that the production biomass was ranged from 0.04 to 0.84 g/L and the total carotenoid from 0.15 to 10.78 mg/L when optimizing culture conditions for *Halorubrum* sp. [25].

## 3. Materials and Methods

### 3.1. Sample Collection and Yeast Isolation

Different sediment samples were collected from the exposed intertidal zone along the South Yellow Sea in Dongtai City, Jiangsu Province, China. Each sediment sample (approximately 100 g) was placed in a sterile plastic bag with an ice bag and transported to the laboratory within 10 h and then processed immediately to isolate yeast. Ten grams of each sample (wet mass) were homogenized in 90 mL sterile 0.9% saline solution then individual yeast colonies were obtained by serial dilution and plating on yeast extract–peptone–dextrose (YPD) agar plates. All the plates were incubated at 28 °C for 24–48 h to determine the morphology of the colony.

### 3.2. Identification of the Red Yeast RY1801 Strain

The pure culture of strain RY1801 was used to investigate its physiological and morphological characteristics according to the methods described by Kurtzman et al. [26]. Genomic DNA of RY1801 was extracted using QIAamp DNA Mini Kit (QIAGEN) following the manufacturer’s instructions. DNA amounts and purity contained in each extract were evaluated by measuring the absorbances at 230, 260 and 280 nm (Nanodrop 2000, Thermo Scientific, Waltham, MA, USA) and calculating the ratio A260/A280 and A260/A230. DNAs were stored at −20 °C prior to use for amplification studies [27]. The ITS region was sequenced using the forward primer ITS1 (5′-TCCGTAGGTGAACCTGCGG-3′) and the reverse primer ITS4 (5′-TCCTCCGCTTATTGATATGC-3′) [28]. The PCR conditions were as follows: 94 °C for 10 min, followed by 30 cycles, 92 °C for 1 min, 52 °C for 1 min, 72 °C for 1 min, and final synthesis at 72 °C for 5 min. The PCR products were separated by agarose gel electrophoresis and purified for sequencing. The sequences obtained were compared to rDNA sequences from the GeneBank (http://www.ncbi.nlm.nih.gov/BLAST/). ITS fragments obtained from GeneBank database were aligned with ClustalW (http://www.ebi.ac.uk/Tools/clustalw2/index.html) and the phylogenetic tree was computed with Jalview 2.4.0.b2 using the neighbor-joining method.

### 3.3. Determination of Biomass and Total Carotenoids

The cells were harvested by centrifugation 8000 rpm and 4 °C for 10 min, later washed with distilled water and centrifuged. The biomass of RY1801 was quantified through drying at 60 °C until a constant mass was obtained.

The carotenoids were extracted using techniques as described by Lopes et al. [29] with slight modification. The 0.1 g dry weight biomass was mixed with 2 mL DMSO and 5 mL acetone in a 10 mL tube. The mixture was subjected to 5 ultrasonic cycles at 40 kHz (Ningbo Scientz Biotechnology Co., Ltd., Ningbo, China) for 10 min, with 5 ml acetone added. The tube was vortexed vigorously and kept standing for 10 min. Centrifugation was performed (5000 × g for 10 min) to remove the biomass from the extracted carotenoids. The biomass was then resuspended in DMSO and acetone for additional extractions. The carotenoids-containing supernatant was pooled and analyzed by spectrophotometry. Initial spectrophotometry scan between 300 and 600 nm revealed the maximum absorption to occur at 490 nm. Carotenoids concentration was determined using the following equation [30,31,32].

Total carotenoids (μg/g of yeast) = A_max_ × D × V/(E × W)

A_max_: the absorbencies of total extract carotenoid at 490 nmD: sample dilution ratioV: volume of extraction solvent (mL)E: extinction coefficient of total carotenoid (0.16)W: dry weight of yeast (g)

### 3.4. Antimicrobial Activity of Carotenoid Pigments

The carotenoid pigments were extracted and dissolved in methanol. The antimicrobial activity was obtained using an agar well diffusion method [33]. *Escherichia coli* ATCC 29522, *Staphylococcus aureus* ATCC 25923 and *Pseudomonas aeruginosa* ATCC27853 were used as indicator strains. After incubation at 37 °C for 24 h, the antimicrobial activity of carotenoid pigments was determined by measuring the diameter of the zone of inhibition.

### 3.5. Optimization of Carotenoid Production in Shake-Flasks Experiments

In order to determine the initial pH values, incubation temperature, carbon source and nitrogen sources on carotenoids production and biomass growth, the experiment was conducted using a series of 250-mL flasks. Each flask contained 100 ml YPD media with 5% inoculum of *Rhodotorula* sp. RY1801. The initial media pH values, incubated temperatures, different carbon sources, and nitrogen sources were adjusted according to the experimental design. The flasks were shaken at 120 rpm for 72 h. The yeast biomass was harvested using refrigerated centrifugation (8000 rpm, 10 min). After washing the cellular pellet with distilled water twice, the biomass was used for further carotenoids extraction and carotenoids production analysis.

## 4. Statistical Analysis

All data were analyzed using One-way Analysis of Variance (ANOVA), and multiple comparison tests (Duncan’s and Tukey’s-tests) were performed using SPSS Statistic 2.0 software. Data were presented as Mean ± Standard. *p* < 0.05 was considered statistically significant.

## 5. Conclusions

Red yeast strain *Rhodotorula* sp. RY1801 was isolated from the exposed intertidal zone along the South Yellow Sea in Dongtai City, Jiangsu Province, China. The optimum conditions found for carotenoids production for *Rhodotorula* sp. RY1801 were as follows: temperature, 28 °C; pH 5.0; carbon source, 10 g/L glucose; and nitrogen source, 10 g/L yeast extract, maximum concentration of 987 µg/L of total carotenoids was obtained. The results of this study showed that the isolated yeast strain *Rhodotorula* sp. RY1801 potentially can be used in the future as promising microorganism for the commercial production of carotenoids.

## Figures and Tables

**Figure 1 marinedrugs-17-00161-f001:**
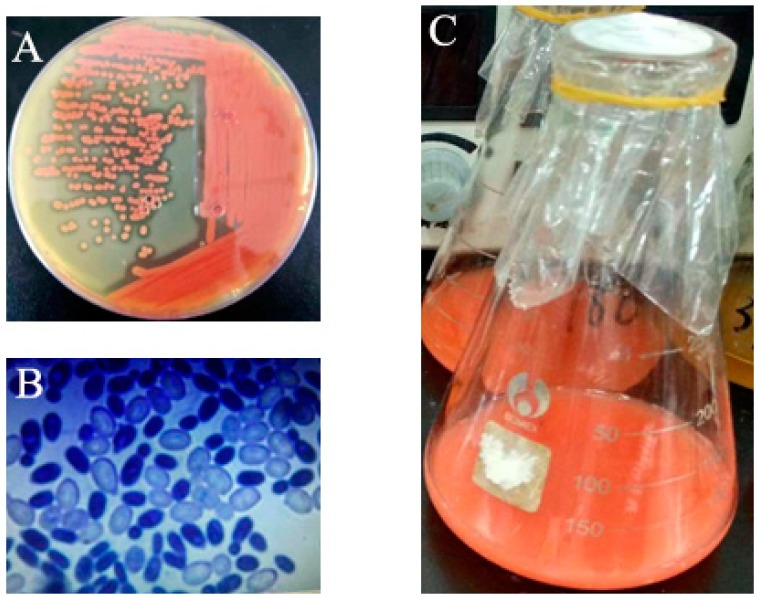
(**A**), Pure culture of the potential marine yeast strain *Rhodotorula* sp. RY1801 on YPD agar. (**B**), Micro-morphology of RY1801 observed under 40× with methylene blue staining. (**C**), Liquid culture of RY1801.

**Figure 2 marinedrugs-17-00161-f002:**
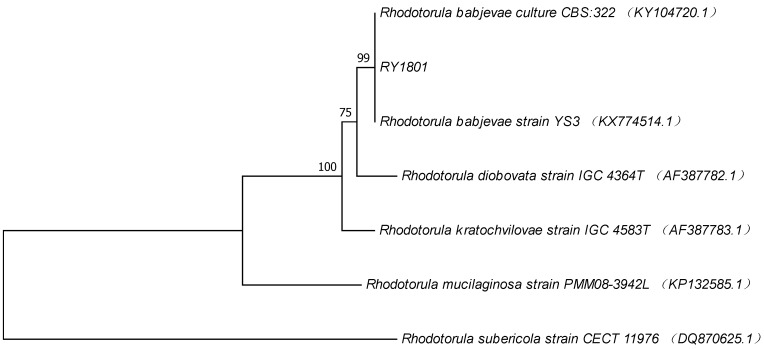
Phylogenetic tree of RY1801 obtained by neighbor-joining analysis of ITS region of rDNA.

**Figure 3 marinedrugs-17-00161-f003:**
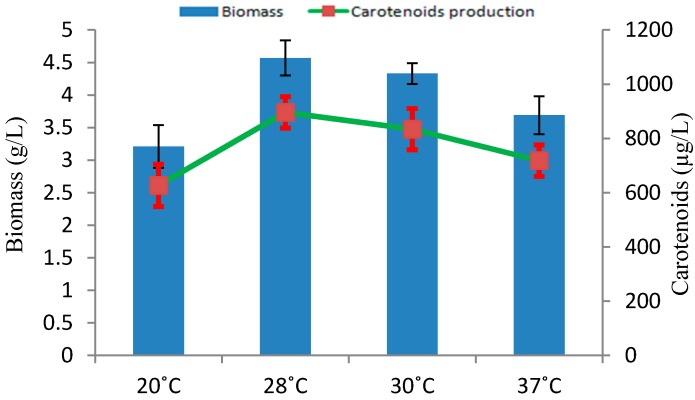
Effect of temperature on biomass and carotenoids production by *Rhodotorula* sp. RY1801.

**Figure 4 marinedrugs-17-00161-f004:**
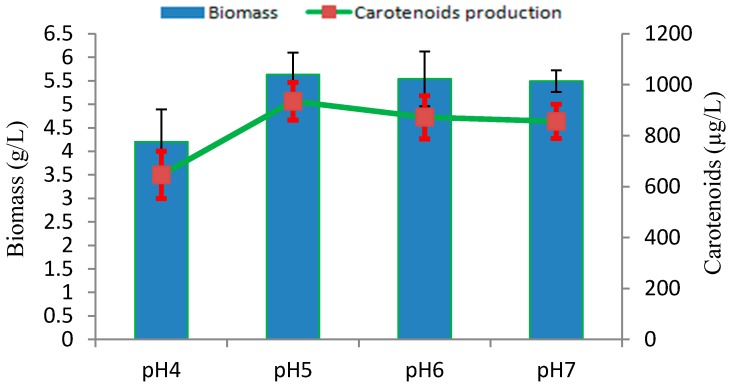
Effect of initial pHs on production of biomass and carotenoids by *Rhodotorula* sp. RY1801.

**Figure 5 marinedrugs-17-00161-f005:**
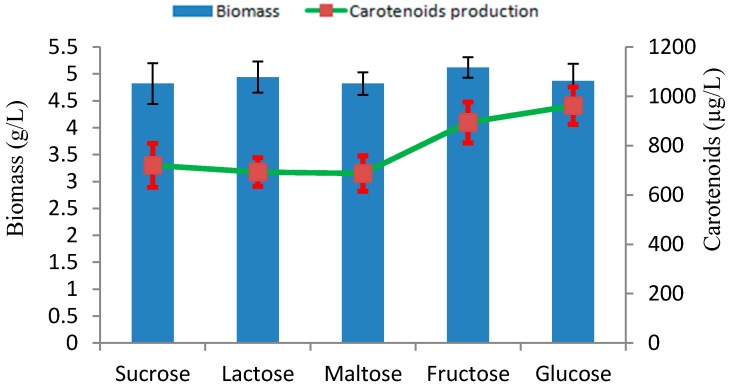
Effect of different carbon sources on production of biomass and carotenoids by *Rhodotorula* sp. RY1801.

**Figure 6 marinedrugs-17-00161-f006:**
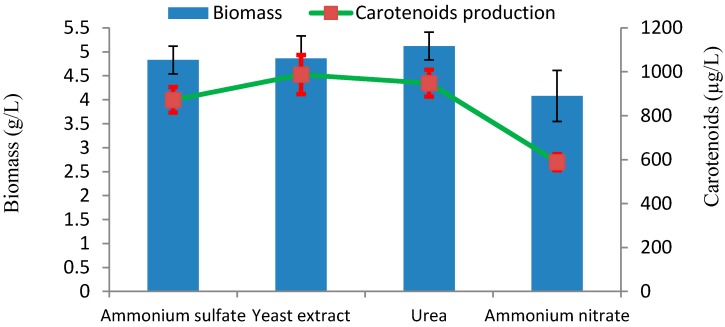
Effect of nitrogen sources on production of biomass and carotenoids by *Rhodotorula* sp. RY1801.

**Table 1 marinedrugs-17-00161-t001:** Morphological, physiological and biochemical characteristics of isolated yeast strain RY1801.

Assimilation Reactions	*Rhodotorula* sp. RY1801	Assimilation Reactions	*Rhodotorula* sp. RY1801	Assimilation Reactions	*Rhodotorula* sp. RY1801
Glucose	+	Ethanol	-	2-keto-d-gluconate	-
Galactose	+	Glycerol	+	Xylitol	-
Sucrose	+	Erythritol	-	50% glucose	-
Maltose	+	Ribitol	+	10% NaCl/5% Glucose	-
Cellobiose	-	Galactitol	+	Starch formation	-
Trehalose	+	d-Mannitol	+	Urease	+
Lactose	-	d-Glucitol	-	Gelatin liquefaction	-
Melibiose	-	*α*-Methyl d-glucose	+	Growth at 19 °C	+
Raffinose	+	Salicin	-	Growth at 25 °C	+
Melezitose	+	d-Gluconate	+	Growth at 37 °C	+
Inulin	-	DL-Lactate	+	Growth at 40 °C	-
Soluble starch	+	Succinate	+	Pellicle	-
d-Xylose	+	Citrate	+	Sedimentation	+
l-Arabinose	+	Inositol	-	True hyphae	-
d–Glucosamine	-	Hexadecane	+	Acid production	-
*N*-acetyl-d-glucosamine	-	Nitrate	+		
Methanol		Vitamin-free

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
