# Peer review of "Isolation, Identification of Carotenoid-Producing Rhodotorula sp. from Marine Environment and Optimization for Carotenoid Production"

_marinedrugs, 2019, doi:10.3390/md17030161_

Reviewer 1 Report

Thee subject of the paper is well suited for this journal even though has been extensively studied over the past years. The authors should provide some additional information and put more emphasis to the novelty and the relevance of their research before publication on Marine Drugs. I am confident that doing so will greatly enhance the impact of this manuscript. Some specific comments, which should be addressed before publication, are listed below.

More specific comments:

The manuscript needs proofreading. There are many editing errors in the text and in the tables, please check carefully the whole  manuscript and correct it.

Abstract:

I found the sentence “A 5% inoculum of Rhodotorula 18 sp. RY1801 was shaken in a 100 ml medium and in a 250 ml flask at 120 rpm for 72 h” too methodological for an abstract.

1. Introduction and discussion

I found the introduction  and discussion too thin, it is suggested the authors to read more recent work.

I think the authors could extract more information from the available literature of the topic, by referring to several papers

Author Response

Dear reviewer:

Thank you for your letters and for the reviewers’ comments concerning our manuscript entitled “Isolation, Identification of Carotenoid-producing Rhodotorula sp. from Marine Environment and Optimization for Carotenoid Production”(ID: marinedrugs-439893). Those comments are all valuable and very helpful for revising and improving our paper, as well as the important guiding significance to our researches. We have studied comments carefully and have made correction which we hope meet with approval. Revised portion are marked in highlight in the paper. The point to point responds to the reviewers’ comments are listed as following:

Responds to the comments:
Comment 1: The manuscript needs proofreading. There are many editing errors in the text and in the tables, please check carefully the whole  manuscript and correct it.
 Response: We have checked the whole manuscript.

Comment 2: I found the sentence “A 5% inoculum of Rhodotorula 18 sp. RY1801 was shaken in a 100 ml medium and in a 250 ml flask at 120 rpm for 72 h” too methodological for an abstract.
 Response: We have deleted the sentence from the abstract.

Comment 3:  I found the introduction  and discussion too thin, it is suggested the authors to read more recent work. I think the authors could extract more information from the available literature of the topic, by referring to several papers.
 Response: We have added some information in the introduction and discussion part.

Reviewer 2 Report

In this paper, authors have reported the isolation and identification of a new Rhodotorula sp. strain which produce carotenoids and it has been optimized. Studies about isolation and identification of pigmented yeasts from natural sources be found in literature (Food and Nutrition Sciences, 2012, 3, 627-633 doi:10.4236/fns.2012.35086; Biosciences Biotechnology Research Asia, 2014, 11:271-278 DOI: 10.13005/bbra/1420)

Even if the paper is understandable, there are some weaknesses that in my opinion make the article not appropriate for publication in Marine Drugs.

1-Line 50. Why only one strain RY1801 was used for further study? What happened with the other 5? Very few details are given about isolation.

2- Line 56 . glactose is written instead of galactose

3- Line 65. Table 1. glactose is written instead of galactose

4- Line 75 to 78 should be in introduction section and not in results and discussion section

5- Line 94. “we found that temperatures higher than 28°C reduced carotenoids synthesis”. Non-significant differences are observed between 28 and 30ºC.

6- In general, in results and discussion section there is a description of the results but accompanied by a poor discussion of them.

7-Line 209. Authors of reference 1 are not well written

Based on all above comments, in my opinion the manuscript does not have the quality standard required to be published in journal Marine Drugs.

Author Response

Dear reviewer:

Thank you for your letters and for the reviewers’ comments concerning our manuscript entitled “Isolation, Identification of Carotenoid-producing Rhodotorula sp. from Marine Environment and Optimization for Carotenoid Production”(ID: marinedrugs-439893). Those comments are all valuable and very helpful for revising and improving our paper, as well as the important guiding significance to our researches. We have studied comments carefully and have made correction which we hope meet with approval. Revised portion are marked in highlight in the paper. The point to point responds to the reviewers’ comments are listed as following:

Responds to the comments:

Comment 1: Line 50. Why only one strain RY1801 was used for further study? What happened with the other 5? Very few details are given about isolation.

Response: Among the six isolates, RY1801 growth faster than others, considering industrial production, we selected the more faster growth strain for further study.

Comment 2: Line 56 . glactose is written instead of galactose.

Response: We have checked it.

Comment 3: Line 65. Table 1. glactose is written instead of galactose.

Response: We have checked it.

Comment 4: Line 75 to 78 should be in introduction section and not in results and discussion section.

Response: We have removed these sentences to the introduction part.

Comment 5: Line 94. “we found that temperatures higher than 28°C reduced carotenoids synthesis”. Non-significant differences are observed between 28 and 30ºC.

Response: We have got the highest carotenoids production at 28°C, and we changed this sentence to “we found that the highest temperature for carotenoids production was at 28°C but with no significant difference when the temperature at 30°C.”

Comment 6: In general, in results and discussion section there is a description of the results but accompanied by a poor discussion of them.

Response: We have added some more discussion.

Comment 7: Line 209. Authors of reference 1 are not well written.

Response: We have checked the reference style.

Reviewer 3 Report

However there are some information that should be clarified in the manuscript (major revisions) in order to be accepted for publication.

The aims of the study should be reformulated according to the research objectives and target results. The conclusions must reflect the innovation of this study and the perspectives.

The text may be handled by a native English speaker.

I didn’t lack new information on other industrial properties of carotenoids (eg. antimicrobial properties). Please describe.

Add a new citation to the introduction and discussion:

Kot, A. M.; Błażejak, S.; Gientka, I.; Kieliszek, M.; Bryś, J. Torulene and torularhodin:“new” fungal carotenoids for industry? Microbial Cell Factories 2018, 17(1), 49. doi:10.1186/s12934-018-0893-z

Why did the authors receive such a small biomass yield? Please compare your results with available scientific literature and describe

Please describe the properties of yeast in food industry (production of carotenoids).

Figures names - yeast strain we write in italics !!! Correct

Author Response

Dear reviewer:

Thank you for your letters and for the reviewers’ comments concerning our manuscript entitled “Isolation, Identification of Carotenoid-producing Rhodotorula sp. from Marine Environment and Optimization for Carotenoid Production”(ID: marinedrugs-439893). Those comments are all valuable and very helpful for revising and improving our paper, as well as the important guiding significance to our researches. We have studied comments carefully and have made correction which we hope meet with approval. Revised portion are marked in highlight in the paper. The point to point responds to the reviewers’ comments are listed as following:

Responds to the comments:

Comment 1: The aims of the study should be reformulated according to the research objectives and target results. The conclusions must reflect the innovation of this study and the perspectives.

Response: We have re-wrote the aims and conclusions.

Comment 2: The text may be handled by a native English speaker.

Response: We have asked a native English speak student for the English revised.

Comment 3: I didn’t lack new information on other industrial properties of carotenoids (eg. antimicrobial properties). Please describe.

Response: We have detected the antimicrobial activity of carotenoid pigments from Rhodotorula sp. RY1801.

Comment 4: Add a new citation to the introduction and discussion: Kot, A. M.; Błażejak, S.; Gientka, I.; Kieliszek, M.; Bryś, J. Torulene and torularhodin:“new” fungal carotenoids for industry? Microbial Cell Factories 2018, 17(1), 49. doi:10.1186/s12934-018-0893-z

Response: We have added this reference in the introduction part.

Comment 5: Why did the authors receive such a small biomass yield? Please compare your results with available scientific literature and describe.

Response: The low biomass yield in our study maybe because of the culture time. In our study, we only cultured the strain for 72 hours (3 days) for carotenoids production. We have discussed in the manuscript.

Comment 6: Please describe the properties of yeast in food industry (production of carotenoids).

Response: Line 44-49, we described the advantage of yeast for carotenoids production.

Comment 7: Figures names - yeast strain we write in italics !!! Correct

Response: We have changed the style.

Round  2

Reviewer 1 Report

In accordance with my comments the authors have provided some additional information from data published earlier, putting more emphasis to the novelty and the relevance of their research.

However the manuscript still needs proofreading. There are still many editing errors in the text and in the tables, please check again carefully the whole manuscript and correct it.
(e.g. line 48 Xanthophylomyces sp..; only 1 . ; line 104 : 20°C without brackets; Table 1 galactose correct Galactose etc…)
1.  Introduction:
In the revised part (line 31 to 34) please add  appropriate references

Author Response

Dear reviewer:

Thank you for your comments concerning our manuscript entitled “Isolation, Identification of Carotenoid-producing Rhodotorula sp. from Marine Environment and Optimization for Carotenoid Production”(ID: marinedrugs-439893). Those comments are all valuable and very helpful for revising and improving our paper. We have made correction according to the comments. Revised portion are marked in highlight in the paper. The point to point responds to the reviewers’ comments are listed as following:

Responds to the comments:

Comment 1: There are still many editing errors in the text and in the tables, please check again carefully the whole manuscript and correct it. (e.g. line 48 Xanthophylomyces sp..; only 1 . ; line 104 : 20°C without brackets; Table 1 galactose correct Galactose etc…)
 Response: We have made some correction to the errors..

Comment 2: Introduction: In the revised part (line 31 to 34) please add appropriate references.

Response: We have added a new reference.

Reviewer 2 Report

In my first report (28-01-2019) I considered that the manuscript did not have the quality standard required to be published in journal Marine Drugs and I proposed this work to be rejected. 

Now I see that authors have considered all comments and suggestion of other reviewers and have improved the quality of the manuscript. I have no objection if this article is accepted to be published in Marine Drugs.

Author Response

Dear reviewer,

Thank you very much for reviewing our manuscript and gave us some construtive opinions.

Best Regards!

Weihua Chu